# OpenReview forum: "Can We Infer Confidential Properties of Training Data from LLMs?"
_NeurIPS.cc/2025/Conference — NeurIPS 2025 spotlight_

### Official Review · Reviewer_3qjY · 2025-06-27

**Clarity:** 3
**Significance:** 3
**Originality:** 4
**Rating:** 5
**Confidence:** 4

**Summary:**

The paper introduces the task of dataset property inference for LLMs. This is specifically targeted at identifying dataset-level confidential information like customer demographics or patient statistics, when LLMs are for instance fine-tuned on domain-specific datasets (healthcare, finance).

Authors identify that prior work has studied a similar question for other data modalities and models, including tabular data and GANs. They argue that property inference for LLMs introduces two new challenges:
- Sensitive properties might be more complex
- LLMs do no fit cleanly into discriminative or generative, make it harder to see what kind of attacks work best

The paper introduces PropInfer, a new benchmark task for property inference in LLMs.
They propose two attacks:
- Prompting the model to generate synthetic samples from the target LLM, to then aggregate them and use the aggregation as an estimated property ratio.
- Shadow-model based attack based on word frequency, training a meta-attack model to predict property ratio based on word frequencies.

They evaluate these attacks in different settings, considering 1 dataset, different properties, multiple pretrained models and two different finetuning methods.

**Questions:**

- In the perplexity attack baseline, how do you compute the perplexity?
- I find the analsysis of which keywords are most predictive quite interesting (Appendix A.3). Could you also add the words that are most useful to make the actual prediction in the best performing meta-attack model? It would be nice to confirm that the model learns that when a word like 'female' is more common, the meta attack model would be more likely to predict a higher female patient ratio.
- For the word frequency attack, do you really need 100k samples? What would happen if you gave the same query budget to the BB attack?
- Maybe a discussion paragraph would be helpful, touching on what other properties would be useful, potentially beyond privacy too. I'm thinking about language distribution, percent of statements biased towards a certain political topic etc.

**Ethical Concerns:**

["NO or VERY MINOR ethics concerns only"]

**Final Justification:**

I liked the paper when I initially reviewed it, and some minor concerns that I raised were addressed during the rebuttal. I'll keep my score (5, accept) and hope the paper gets accepted.

**Limitations:**

Only one dataset, but this is not a major concern.

**Paper Formatting Concerns:**

Line 53, is it reflect rather than reflects?

**Quality:**

3

**Strengths And Weaknesses:**

**Strengths:**
- The paper introduces a novel vulnerability and attack in the context of LLMs, the relevance of which is clearly articulated throughout the work.
- The dataset considered in the paper very nicely fits the purpose of the attack.
- Considering both SFT and CLM is interesting to study from a memorization perspective, also in the context of gender (more in the question) and diagnosis (more in the answer).
- The two attacks proposed seem to work quite well, beating well thought-through baselines.
- Overall, I really like the problem the paper studies and believe the experiments are adequate evidence for the claims made.

**Weaknesses:**
- I think the paper could benefit from a bit more related work on other (privacy) attacks leveraging similar techniques. For instance, [1,2] use word frequencies or an n-gram model to run black-box access MIAs, with [2] also doing shadow modeling. While it's quite different, [3] might be useful to.
- Only one dataset is used to evaluate the methods, but I would not call this a major concern. I would just find it interesting to see if authors find other use-cases where this is problem statement is equally relevant and the attacks are equally successful.

[1] Kaneko, M., Ma, Y., Wata, Y., & Okazaki, N. (2024). Sampling-based pseudo-likelihood for membership inference attacks. arXiv preprint arXiv:2404.11262.

[2] Meeus, M., Wutschitz, L., Zanella-Béguelin, S., Tople, S., & Shokri, R. (2025). The Canary's Echo: Auditing Privacy Risks of LLM-Generated Synthetic Text. arXiv preprint arXiv:2502.14921.

[3] Hayase, J., Liu, A., Choi, Y., Oh, S., & Smith, N. A. (2024). Data Mixture Inference: What do BPE Tokenizers Reveal about their Training Data?. arXiv preprint arXiv:2407.16607.

---

> ### Author Rebuttal · Authors · 2025-07-31
>
> We sincerely thank the reviewer for taking the time to review our paper and provide thoughtful comments. We provide our responses to your concerns and questions below.
>
> **MIA leveraging word frequencies in literature (W1):** Thank you for highlighting these related works on membership inference attacks (MIA). They complement our findings and help explain why word frequencies or n-grams are strong signals in the context of LLMs. Consistent with prior work, we also observe that loss-based scores—effective in MIA and property inference for earlier models like ConvNets and linear classifiers—are less informative for LLMs. We will incorporate this discussion in the revised version.
>
> **Details for perplexity attack baseline (Q1)**: The perplexity attack baseline follows the set-up in [1]. We have two hold-out dataset $S_0$ and $S_1$ where the property ratios are 0% and 100%, respectively. Then we calculate the average perplexity scores for each dataset for each model $f$, yielding a two-dimensional vector that serves as the shadow feature. The perplexity attack baseline defines the mapping from the model $f$ to this two-dimensional vector as the shadow feature function. These details are presented in the Appendix A.2 and we will move these details to the main paper for better understanding this baseline!
>
> [1] A. Suri and D. Evans. Formalizing and estimating distribution inference risks, 2022.
>
> **More details about the keywords in word-frequency attack (Q2):** Yes, we are happy to provide additional details on how keyword frequencies correlate with the property ratio in the fine-tuning dataset. Please see our response **“Intuition of word-frequency signals on L230–231” in the reply to Reviewer nLBK.** For example, when the target dataset has a higher Female ratio, the word "female" appears more frequently in the outputs. We also report the list of selected key words from the shadow model with the corresponding correlation coefficients with the ratio in that response.
>
> **Word-frequency attack needs more samples than BB generation attack (Q3):** We would like to answer two questions separately.
>
> 1. We have studied the impact of query budget on the BB generation attack. As shown in Figure 7 (Appendix), the performance for LLaMA-1 and Pythia converges around 2k samples. Therefore, increasing the query budget to 100k is unlikely to significantly improve the results.
>
> 2. Why does word-frequency attack need more than $100k$ samples, much more than the BB generation attack? Unlike the BB generation attack, which directly predicts the property ratio, the word-frequency attack relies on the frequency of specific keywords—many of which occur much less frequently than the property ratio itself. For instance, when the target female ratio is 0.7, the word "female" appears in only 2.5% of samples (the female property can be indicated by other words too). A query budget of 2k introduces an estimation error of around 0.01, which may still allow the BB attack to accurately predict a ratio close to 0.7 (e.g., 0.69). However, this level of noise significantly disrupts the shadow model's ability to learn from small word-frequency variations—e.g., "female" frequencies are 0.016, 0.0219, and 0.025 for target ratios 0.3, 0.5, and 0.7, respectively. Many other informative keywords have even lower frequencies, making the attack highly sensitive to sample size. This necessitates a larger query budget to preserve the correlation signals needed for effective property inference.
>
> **Other confidential properties (Q4, W2):** Thanks for bringing up the new practical examples. By considering a dataset of political news and the property of any political topic, the property inference from a news model that is finetuned on such dataset can be another practical scenario. We will add this discussion in the revision!

---

### Official Review · Reviewer_nLBK · 2025-07-03

**Clarity:** 4
**Significance:** 3
**Originality:** 4
**Rating:** 5
**Confidence:** 3

**Summary:**

This paper tackles the dataset property inference task in LLMs. Similar to membership inference, evaluating dataset property inference helps to protect confidential attributes of datasets from leaking. Given lack of property evaluation setup in LLMs, this paper introduces PropInfer which leverages the ChatDoctor dataset to construct two setups matching two LLM fine-tuning paradigms. In addition, the paper illustrates the effectiveness of various attacking methods including a novel one tailored for LLM. The results suggest that property inference is a critical issue in LLMs that requires attention to better defense.

**Questions:**

1. While I appreciate the Generation w/o FT baseline, it is still unclear what effects pretraining has on the property inference. Particularly for larger models, existing biases (e.g., gender bias in doctor-patient setting) from large-scale pretraining may interfere with property inference evaluation. I would love to see this being discussed if not experimented.
2. Continuing from (1), the set of models evaluated is limited given the author claims PropInfer as a benchmark. It would be really assertive and make the paper stronger if more models (particular larger ones provided your resource permits) are included in the experiments. It can help clarify influence from large-scale pretraining and showcase generalization of the method to most LLMs.
3. The author mentioned an intuition on L230-231. However, this hypothesis is not well-tested. Later on L338-339 and in Appx A.3, it also seems that the performance is sensitive to the choice of $d$. This rationale deserves a deeper dive to both understand the paper better and make the method more convincing.
4. What is the source for the claim on L93-95? Even if the claim is true, it is still worth to add as a baseline and show that it is less effective than proposed attacks.
5. Appx A.1 mentioned a human study. Could you provide the survey details? E.g., how many participants and are they medical experts or general audience?

I would raise my score if the experiment results can be verified more generally and/or some of the concerns/hypothesis/claims above can be clarified.

**Ethical Concerns:**

["NO or VERY MINOR ethics concerns only"]

**Final Justification:**

The rebuttal addresses most of my concerns. I still think that comparisons among more models should be included as a benchmark, but I do understand the limited time authors have during the rebuttal period. It would be great to include more in the camera ready. After considering comments from other reviewers, overall, I'll raise my score to Accept.

**Limitations:**

There is one limitation which is tied to Question 1 above. Given the social biases in pretrained large-scale models, it may influence the property inference results and mistakenly inherit the bias as dataset property.

**Paper Formatting Concerns:**

A typo on L334: I'm guessing it's missing a figure reference.

**Quality:**

3

**Strengths And Weaknesses:**

Strength:
1. The paper is well motivated. The presentation is clear and effective.
2. The benchmark setup is convincing, comprehensive and robust.
3. The result and analysis is reasonable and demonstrates effectiveness.

Weakness (detailed in Questions section below):
1. As a benchmark, the result section is somewhat limited in terms of variabilities in model architecture, size, etc. (See Q2 below).
2. While most of the claims and motivations are well-supported. Some important claims can be further clarified (See Q3&4 below).

---

> ### Author Rebuttal · Authors · 2025-07-31
>
> We sincerely thank the reviewer for taking the time to review our paper and provide thoughtful comments! We provide our responses to your concerns and questions below.
>
> **The effect of bias from pre-training on property inference (Q1):** Yes, intuitively, pre-training bias may affect the effectiveness of property inference, depending on both the pre-trained model and the target property ratio. In our experiments, we evaluated multiple pre-trained models across various gender ratios. We first examined the ratio of the target property (female) indicated by the pre-trained models, as shown in the following table:
>
> | Model     | LLaMA-1 | Pythia | LLaMA-3 |
> |-----------|---------|--------|---------|
> | Pre-train ratio of Female | 64.2%  | 67.7% | 62.6%  |
>
> The pre-train female ratios among all three models are between 60%-70%. We check the results in Table 1 and have the following observations:
>
> 1. In Chat-Completion mode, both the BB generation attacks and shadow model attack perform effectively across target female ratios ranging from 30% to 70%.
>
> 2. In Q&A mode, we observe that the BB generation attack yields significantly higher error at a target female ratio of 30%, which has the greatest deviation from the pre-train ratio, compared to the errors observed at 50% or 70%. On the contrary, the shadow model attack based on word-frequency is more effective in the QA mode.
>
> These observations suggest that (1) In Chat-Completion mode, the models learn the full distribution of the fine-tuning data and pre-training bias has minimal impact on property inference. (2) In Q&A mode, where fine-tuning does not optimize over full text but only answers, pre-training bias may have a greater influence on subsequent property inference. Additionally, the shadow-model attack with word-frequency has stronger ability than BB generation attack to overcome this bias.
>
> **Evaluating more models (Q2, W1):** We evaluate our PropInfer benchmark on the Qwen-2.5-7B-Instruct model and observe similar trends. The attack results for the **QA mode (MAE)** are shown below.
>
> **Gender property**
> | Method           | Target Female Ratio = 30 | 50            | 70             |
> |------------------|---------------------------|---------------|----------------|
> | BB generation    | 16.7 ± 3.54               | 2.71 ± 1.68   | 20.17 ± 4.10   |
> | Perplexity       | 30.3 ± 4.97               | 17.71 ± 9.35  | 7.87 ± 3.03    |
> | Word-Frequency   | 0.812 ± 0.45              | 5.12 ± 4.08   | 13.09 ± 18.2   |
>
> **Medical diagnosis property**
> |                  | Mental        | Digestive     | Childbirth     |
> |------------------|---------------|---------------|----------------|
> | w/o fine-tuning  | 3.92          | 6.6           | 10.55          |
> | BB generation    | 0.95 ± 0.63   | 2.93 ± 0.21   | 7.56 ± 0.41    |
>
> The observations on Qwen-2.5 are consistent with those from other models presented in the paper. For example, in the case of the Female property—which is more likely to appear in the question (input)—the shadow model using word-frequency features performs best when the model is fine-tuned in QA mode. In contrast, for medical diagnosis properties, which are more commonly found in the answer (output), the BB generation attack achieves strong performance. We will include the full results in the revised version.
>
> **Intuition of word-frequency signals on L230-231 (Q3-1, W2):** We begin by examining how word frequencies vary with different female ratios in the fine-tuning dataset. The tables below show the number of occurrences of selected words across 150k i.i.d. samples generated by the fine-tuned LLaMA-1 model under varying target female ratios.
>
> | Target female ratio | 0.3  | 0.5  | 0.7  |
> |---------------------|------|------|------|
> | "female"            | 2484 | 3296 | 3775 |
> | "spotting"          | 119  | 149  | 205  |
>
> We observe that the occurrences of both words increase monotonically with the target female ratio.
>
> Furthermore, for each word, we compute the correlation coefficient between its occurrence frequency and the corresponding female ratio. We present the list of selected key words from the shadow model with the corresponding correlation coefficients with the ratio.
>
> -- Model: Llama-1 (CC mode): The top 10 keywords are (his, himself, her, he, female, him, prostate, she, son, daughter) and the correlation coefficients of −0.956, −0.934, 0.947, −0.950, 0.960, −0.933, −0.931, 0.954, −0.929, and 0.932, respectively.
>
> -- Model: Pythia (CC mode): The top 10 selected keywords are (scrotum, he, penis, foreskin, male, glans, female, masturbate, masturbation, tip) and the correlation coefficients of −0.941, −0.916, −0.944, −0.934, −0.928, −0.940, 0.968, −0.919, −0.921, and −0.917, respectively.
>
> -- Model: Llama-3 (CC mode): The top 10 keywords are (penile, erect, penis, scrotum, female, scrotal, males, masturbating, erection, erectile) and the correlation coefficients of −0.940, −0.935, −0.939, −0.935, 0.980, −0.941, −0.935, −0.945, −0.945, and −0.934, respectively.
>
> -- Model: Llama-1 (QA mode): The top 5 selected keywords are (female, reluctant, spotting, scanty, ovaries) and the correlation coefficients are 0.785, 0.774, 0.787, 0.821, and 0.782, respectively.
>
> -- Model: Pythia (QA mode): The top 10 selected keywords are (pelvic, recurring, football, bland, indigestion, bothering, uti, point, presenting, smear) and the correlation coefficients of 0.738, 0.730, −0.740, 0.760, 0.740, 0.822, 0.746, 0.743, −0.751, and 0.753, respectively.
>
> -- Model: Llama-3 (QA mode): The top 10 selected keywords are (nifedipine, readings, yielding, squats, analogs, smoke, particular, cigarette, quit, regularly) and the correlation coefficients are −0.786, −0.788, −0.765, −0.788, −0.791, −0.773, 0.847, −0.774, −0.810, and −0.832, respectively.
>
> A correlation coefficient close to 0 indicates no correlation, while values near 1 or -1 indicate strong positive or negative correlation, respectively. We find that many selected keywords exhibit strong correlations with the target female ratio, supporting the effectiveness of word frequency as a signal. Notably, the correlations are generally stronger when the target model is in CC mode compared to QA mode—aligning with the performance differences observed in Table 1, where word-frequency attacks perform better under CC mode than QA mode.
>
> **The choice of the number of key words d (Q3-2, W2):**  When training the meta attack model $g$, the parameter $d$ represents the number of features. From a learning perspective: when $d$ is too small, the features are not sufficient enough to predict the label (target ratio) correctly; conversely, when $d$ is too large, the limited number of shadow models—due to the high cost of training them—may lead to overfitting and does not learn useful mapping that generalize well.
>
> **The relationship between property inference attack and data extraction attack (Q4, W2)** As noted in [1], data extraction attacks on large language models tend to recover training examples that appear multiple times, introducing a repetition bias. This means the extracted data may not represent the full training distribution, especially for rare or non-repetitive samples. This highlights a key distinction between data extraction and property inference: while data extraction is considered successful if any training samples are recovered, property inference aims to recover global properties of the training set. Therefore, strong performance in data extraction does not imply effectiveness in property inference, which motivates our decision to study property inference as a distinct task.
>
> [1] Carlini, Nicholas, et al. "Extracting training data from large language models." 30th USENIX security symposium (USENIX Security 21). 2021.
>
> **Human study (Q5):** Besides the authors, we invite one general audience to help evaluate the labeling performance of ChatGPT.

---

> > ### Comment · Reviewer_nLBK · 2025-08-06
> >
> > Thank you for the detailed explanation and additional experiments. The rebuttal addresses most of my concerns. I still think that comparisons among more models should be included as a benchmark, but I do understand the limited time authors have during the rebuttal period. It would be great to include more in the camera ready. After considering comments from other reviewers, overall, I'll raise my score to Accept. Thanks!

---

> > > ### Author Response · Authors · 2025-08-06
> > >
> > > Thank you for raising the score to Accept! We greatly appreciate your positive feedback and support of our work.

---

### Official Review · Reviewer_g3oc · 2025-07-03

**Clarity:** 3
**Significance:** 3
**Originality:** 3
**Rating:** 5
**Confidence:** 4

**Summary:**

This paper studies property inference attacks on large language models (LLMs). Specifically, it asks whether adversaries can recover dataset-level attributes rom an LLM fine-tuned on sensitive data. The authors introduce a benchmark task PropInfer, based on the Chat-Doctor medical Q&A dataset. They consider two fine-tuning modes (Q&A and chat completion) and select a range of sensitive properties (e.g. disease rates, gender balance). They propose two attack strategies: (1) a black-box generation-based attack that crafts prompts and measures the frequency of outputs containing the target property; and (2) a shadow-model approach that trains auxiliary models with known property ratios, extracts word-frequency statistics, and uses a meta-classifier to infer the target ratio. Experiments show that both attacks can successfully estimate dataset attributes in many cases. For instance, the shadow attack is highly effective when the model is fine-tuned in Q&A mode and the property is manifest in questions, whereas the generation attack works well in completion mode or when attributes appear in both questions and answers. The results reveal a nontrivial vulnerability that black-box LLM access can leak aggregate statistics of the fine-tuning data.

**Questions:**

Figure 1. Why does "daughter" indicates female? Could it also be male?
For black box attacks on sentence completion, how many prompts and samples are generated to get a reliable estimate? How sensitive are attacks to changes in wording of prompts?
For the shadow attack, how large was the auxiliary dataset and how was the keyword list selected?

**Ethical Concerns:**

["NO or VERY MINOR ethics concerns only"]

**Final Justification:**

The authors have addressed the comments and I adjusted the rating from borderline accept to accept.

**Limitations:**

Yes

**Paper Formatting Concerns:**

Overall, formatting looks good and compliant to Neurips guidelines.
Some minor typos (e.g. “infer confidential” should be “infer confidential properties”).
Redundant sentence line 45 - 47  "There are two standard ways to fine-tune an LLM with this dataset that correspond to two use-cases:
question-answering and chat-completion. According to the use case, our benchmark task has two
modes where the models are fine-tuned differently – Q&A Mode and Chat-Completion Mode."

**Quality:**

3

**Strengths And Weaknesses:**

The paper expands on prior works on property inference and apply to LLMs.
Authors defined two modes and used Chat-Doctor to infer on sensitive aggregate stat which felt relevant to real world scenario. Overall, I felt the  evaluation is thorough and strong. The authors evaluate multiple scenarios and baselines, demonstrating clear cases where private proportions can be predicted. The results are well-analyzed and provides actionable insights.

One thing I wish the authors addressed is to delve into why childbirth rate performed poorly vs other properties. Also, showing whether these attacks would generalize to other domains would be valuable.
Additionally, I wish the authors expanded more on  potential defenses. The paper establishes the attack side but does not explore how one might mitigate these leaks.

---

> ### Author Rebuttal · Authors · 2025-07-31
>
> We sincerely thank the reviewer for taking the time to review our paper and provide thoughtful comments! We provide our responses to your concerns and questions below.
>
> **Poor performance to infer childbirth rate (W1):** We would like to elaborate more details to explain the relatively poorer performance for the childbirth ratio than the other properties. We checked **the exact ratios of three medical properties** from the pre-trained models, which are presented in the following table:
> |                          | Mental | Digestive | Childbirth |
> |--------------------------|:------:|:---------:|:----------:|
> | Pre-train ratio in LLaMA-1 | 1.65%  |  8.48%    |   0.71%    |
> | Pre-train ratio in Pythia  | 3.67%  |  3.11%    |   0.75%    |
> | Pre-train ratio in LLaMA-3 | 1.65%  |  8.03%    |   0.279%   |
>
> We observe that the pre-train ratio for Childbirth is notably lower than the pre-train ratio for the other two properties. This may help explain the weaker property inference performance on fine-tuned models for Childbirth: the pre-trained models may not have enough prior knowledge about Childbirth property (possibly due to some safety training that removes the sensitive topics such as childbirth); hence, the model may struggle to effectively learn Childbirth-related content during fine-tuning, making property inference for this attribute more difficult.
>
> **Potential defenses (W2):** we agree that evaluating the attack under potential defenses is valuable. As part of the rebuttal, we implemented a simple defense: since the attack depends on model generations, we adjusted the temperature parameter $T$ in the final softmax layer. Note that $T>1$ makes the model 's output more balanced among all tokens and $T<1$ makes the model's output more concentrated on the high-probability tokens.
>
> We empirically evaluate this defense on the BB-generation attack using LLaMA-3 fine-tuned in CC mode. The table below reports **the average predicted ratios** inferred by the attack under different temperature settings, given fine-tuning datasets with varying female ratios.
>
> |                            |   T = 1 (no defense) |   T = 0.5   |   T = 1.5   |   T = 3.5   |
> |-----------------------------|:--------------------:|:-----------:|:-----------:|:-----------:|
> | Target Female Ratio = 0.3   |        0.355         |    0.200    |    0.415    |   0.4585    |
> | Target Female Ratio = 0.5  |        0.516         |    0.447    |    0.542    |   0.5657    |
> | Target Female Ratio = 0.7 |        0.6958        |   0.8686    |   0.6453    |   0.6866    |
>
> We observe that without defense ($T=1$), the attack predict the target female ratio mostly correctly. When the temperature is altered $(T \neq 1)$, the predicted ratios have larger error, indicating that adjusting decoding settings can serve as a simple yet effective defense against sampling-based attacks. However, such a defense may break down if the adversary is aware of the default temperature setting or has access to the model weights. A deeper investigation into the limitations of this defense, as well as the development of more robust mitigation strategies, is an important direction for future work. We will include the additional results and the above discussion in our revised version.
>
> **Clarification on Figure 1 (Q1):** The “Female” property refers to the gender of the patient. In Figure 1, the patient is the daughter, as indicated by the parent seeking medical advice on her behalf. We will revise the text to define the “Female” property more clearly and avoid confusion.
>
> **Clarification on sample sizes for black-box attacks (Q2):** Figure 2(c) in the main paper shows the ablation study on the effect of sample sizes on the performance of BB generation attack when targeting LLaMA-3 on the gender property, and Figure 7 in the appendix shows the ablation study on other models. We can observe that the attack performance converges around 2k samples for LLaMA-1 and Pythia, and shows near-convergence for LLaMA-3.
>
> **Study on the sensitivity of prompt wording and the number of prompts (Q3):** We conduct an ablation studied on the effect of individual prompts on the performance of BB-generation attack in Table 3 (appendix). The results show that while there is some variance in attack performance across different prompts, the attack remains consistently effective—indicating that its success is not sensitive to prompt choice. In addition, we observe that aggregating three prompts is better than a single prompt for most cases across different models and different target ratios.
>
>
> **Clarification on shadow attack (Q2):** The auxiliary dataset has 14791 data, enabling the subsampling of size 6500 (same as fine-tune size) with different desired ratios. As for selecting the keywords, we first construct an $k_1k_2\times |V|$ feature matrix $X$ where $X_{i, j}$ indicate the $j$th word’s frequency in the $i$th shadow model, and a label vector $y$, where $y_i$ indicates the property ratio of the $i$th shadow model. Then we apply the F-regression (https://scikit-learn.org/stable/modules/generated/sklearn.feature_selection.f_regression.html) for this regression task (X, Y), which assigns the score to each feature. Then we are able pick the top-$d$ features (i.e. words) according to the scores. We will add more details in the revision about this feature selection procedure.

---

### Official Review · Reviewer_v2D5 · 2025-07-03

**Clarity:** 2
**Significance:** 3
**Originality:** 3
**Rating:** 4
**Confidence:** 3

**Summary:**

This paper investigates property inference attacks on fine-tuning datasets of large language models (LLMs). The authors identify a gap in existing research, which has primarily focused on property inference attacks against discriminative models (e.g., image classification) and generative models (e.g., GANs), while overlooking LLMs. To address this gap, they propose a prompt-based generation attack and a shadow-model attack that leverages word frequency signals. They evaluate their approaches on the ChatDoctor dataset to show the effectiveness of their design.

**Questions:**

1. Could the authors provide stronger justification for the gray-box assumptions?

2. Could the authors provide more detailed explanations of the core attack mechanisms?  Specifically, how are prompts constructed for the generation-based attack? What is the rationale for using word frequency features in the shadow-model attack, and how does this choice compare to alternative feature representations?

3. Could the authors extend the evaluation to include additional datasets beyond ChatDoctor and more diverse property types?

4. Can the proposed methods be adapted to infer properties of pre-training datasets, rather than just fine-tuning datasets?

**Ethical Concerns:**

["NO or VERY MINOR ethics concerns only"]

**Final Justification:**

The authors have addressed my concerns to some extent. However, I still have reservations regarding the evaluation and its comprehensiveness. Despite these remaining concerns, the clarifications provided are sufficient for me to increase my rating to 4.

**Limitations:**

yes

**Paper Formatting Concerns:**

/

**Quality:**

3

**Strengths And Weaknesses:**

Strengths
1. The paper studies an interesting problem.
2. The proposed attacks are specifically designed for LLMs and provide tailored property inference tools.
3. The experimental results demonstrate the effectiveness of the proposed attacks and provide insightful observations.

Weaknesses
1. The gray-box setting assumes that the attacker has knowledge of the pre-trained model, fine-tuning method, instruction template, and target dataset size. This seems unrealistic for real-world scenarios where attackers typically only have API access without detailed training information. The assumption of knowing the exact dataset size is particularly strong. The authors may justify the assumptions.

2. The paper lacks clear explanations of key technical details. The paper does not clearly explain how prompts are constructed for the generation-based attack, and why word frequency was chosen as the feature for the shadow-model attack. Additionally, it is unclear why existing shadow-model attacks cannot be directly applied to this setting.

3. The experimental scope is limited. The evaluation is limited to a single dataset (ChatDoctor) and only two property types (gender and diagnosis). The authors may consider additional datasets and property types.

4. The proposed methods target only fine-tuning datasets, potentially overlooking the more significant threat of attacking pre-training datasets, which may contain larger volumes of sensitive information.

---

> ### Author Rebuttal · Authors · 2025-07-31
>
> We sincerely thank the reviewer for taking the time to review our paper and provide thoughtful comments! We provide our responses to your concerns and questions below.
>
> **Justification for the gray-box assumptions (W1, Q1):** The gray-box setting we adopt is standard in prior work on attacks like membership inference [1, 2] and is intended to assess the upper bound of a realistic adversary’s capability. We agree that relaxing these assumptions offers deeper insight into the practical effectiveness of the attack; hence, in our rebuttal, we further examine assumptions about access to the pre-trained model and target dataset size.
>
> To test the assumption of access to the same pre-trained model, we evaluate a setting where the adversary uses LLaMA-3 for training shadow models but targets a LLaMA-1 model. MAE results for both CC and QA modes are shown in the table below.
>
> **CC mode (MAE)**
> | Target female ratio(%)        |   30    |   50   |   70    |
> |-------------------------------|:-------:|:-------:|:-------:|
> | Knowing the pre-trained model |  3.44%  |  0.27%  |  6.26%  |
> | Without knowing the model     |  3.85%  |  0.17%  | 15.31%  |
>
> **QA mode (MAE)**
> | Target female ratio(%)        |   30    |   50    |   70    |
> |-------------------------------|:-------:|:-------:|:-------:|
> | Knowing the pre-trained model |  6.86%  | 12.96%  |  6.82%  |
> | Without knowing the model     | 20.23%  |  0.24%  | 20.00%  |
>
> We observe that the knowledge of the pre-trained model has small effects in CC mode, but notable effects in the QA mode. This might be because with the CC mode, the model is learned to fully mimic the texts in the fine-tuning dataset, disregarding what the pre-trained model is. However in QA mode, while the model learns to answer questions based on the fine-tuning data, the generation of the questions themselves may still rely heavily on the pre-trained model, leading to greater divergence in the resulting question–answer pairs.
>
> To validate the dataset size, we additionally test our word-frequency attack by training shadow models with the dataset size 3000 – the ground truth fine-tuning dataset size is 6500. The inference results (MAE) towards the target Llama-1 model trained with CC mode are reported below:
>
> | Target Female Ratio = (%)      |   30    |   50    |   70    |
> |-------------------------------|:-------:|:-------:|:-------:|
> | Knowing the fine-tuning size  |  3.44%  |  0.27%  |  6.26%  |
> | Without knowing the size      |  3.75%  |   0%    |  10.0%  |
>
> Similar to the observation with pre-trained model architecture, we find that knowledge of the fine-tuning dataset size has minimal impact on the performance of the property inference attack. This may be because with the CC mode, the model is trained to closely mimic the fine-tuning data, so even if the fine-tuning dataset size varies, the model’s output distribution remains consistent.
>
> We will add the results to the revision as an analytical study for our word-frequency attack!
>
> [1] N. Carlini, S. Chien, M. Nasr, S. Song, A. Terzis, and F. Tramer. Membership inference attacks from first principles. In 2022 IEEE symposium on security and privacy (SP), pages 1897–1914. IEEE, 2022.
>
> [2] Watson, Lauren, et al. "On the importance of difficulty calibration in membership inference attacks." arXiv preprint arXiv:2111.08440 (2021).
>
> **Details about the prompt construction (Q2-1, W2-1)**: The prompts used in both the BB-generation and word-frequency-based attacks are listed in Appendix A.4. We choose three generic prompts, such as “Hi, ChatDoctor, I have a medical question.” without any prompt engineering or cherry picking.
>
> We conduct an ablation study on the effect of individual prompts on the BB-generation attack: as shown in (Table 3 Appendix), we observe that while there is some variance in attack performance across different prompts, the attack remains consistently effective—indicating that its success is not sensitive to prompt choice.
>
> **The word-frequency feature versus existing features such as loss or perplexity (Q2-2, W2-2)**: Our perplexity-based baseline adapts the existing shadow model attack from [1] to the LLM setting by computing average perplexity over two hold-out datasets with 0% and 100% property ratios, forming a two-dimensional shadow feature (see Appendix A.2 for details). As shown in the results (Table 1), this baseline is consistently outperformed by our shadow-model attack based on word-frequency, indicating that perplexity is not a strong signal for property inference in LLMs.
>
> We provide additional justification for using word frequency as a feature for the shadow model attack (see our response on Intuition of word-frequency signals on L230-231 to Reviewer nLBK). To support this, we report correlations between selected keywords and property ratios to test the hypothesis that certain properties are closely linked to specific word frequency in the generated text.
>
> [1] A. Suri and D. Evans. Formalizing and estimating distribution inference risks, 2022.
>
> **Other potential dataset (Q3, W3)**: In our experiments with the ChatDoctor dataset, we varied three gender ratios and selected three additional medical attributes as targets for property inference. Meanwhile, we would like to discuss other practical scenarios which might support the evaluation – such as the one suggested by Reviewer 3qjY—where a model fine-tuned on political news might reveal the distribution of political topics in its training data.
>
> **The extension to property inference on pre-trained models (Q4, W4)**: Property inference on fine-tuned models has its own practical scenarios as fine-tuning datasets are often application- and user-specific, containing sensitive properties that should remain private. While we agree that the property inference on pre-trained models represents an interesting threat model, studying this setting requires either access to ground truth properties of existing public LLMs or substantial computational resources to pre-train models from scratch on controlled datasets–making it computationally infeasible for us.

---

> > ### Comment · Reviewer_v2D5 · 2025-08-05
> >
> > Thank you for conducting the additional experiments and providing clarifications. To further strengthen the evaluation, it would be valuable to assess the proposed method on the potential dataset mentioned by the authors

---

### Decision · Program_Chairs · 2025-09-17

**Decision:**

Accept (spotlight)

**Comment:**

This paper is concerned with property inference attacks on finetuning datasets on LLMs. The paper creates a benchmark for valuating property inference attacks suspeptability and introduce two new property inference attacks tailored to LLMs - prompt-based generation attack and shadow-model attack. Empirical results demonstrate empirical success

This paper studies Interesting and overlooked (in context of LLMs) problem and therefore is novel and impactful. The paper is well written, the empirical evidence is compelling, the experimental results and baselines are comprehensive and well thought through and the results demonstrate vulnerabilities of model LLMs to these types of attacks.

Authors did an excellent job providing additional insights while relaxing some of the original assumptions (using the same model, reliance on dataset size etc). Additionally authors proposed during the rebuttal a simple defense for their attacks and demonstrated its performance. We hope these additional results will be incorporated into the final version of the paper.

On the areas of improvement, reviewers recommend exploring additional datasets and models and considering attacks on pretraining datasets as opposed to finetuning ones as currently done.